# Bioinspired and Post-Functionalized 3D-Printed Surfaces with Parahydrophobic Properties

**DOI:** 10.3390/biomimetics6040071

**Published:** 2021-12-13

**Authors:** Léna Ciffréo, Claire Marchand, Caroline R. Szczepanski, Marie-Gabrielle Medici, Guilhem Godeau

**Affiliations:** 1Institut Méditerranéen du Risque de l’Environnement et du Développement Durable (IMREDD), Université Côte d’Azur, 06200 Nice, France; lena.ciffreo@etu.univ-cotedazur.fr (L.C.); claire.marchand@etu.univ-cotedazur.fr (C.M.); 2Department of Chemical Engineering & Materials Science, Michigan State University, East Lansing, MI 48824, USA; szcz@msu.edu; 3Institut de Physique de Nice (INPHYNI), Université Côte d’Azur, UMR 7010, 06000 Nice, France; Marie-Gabrielle.MEDICI@unice.fr

**Keywords:** plants, bioinspiration, parahydrophobic, harvesting, 3D printing, post functionalization

## Abstract

Desertification is a growing risk for humanity. Studies show that water access will be the leading cause of massive migration in the future. For this reason, significant research efforts are devoted to identifying new sources of water. Among this work, one of the more interesting strategies takes advantage of atmospheric non-liquid water using water harvesting. Various strategies exist to harvest water, but many suffer from low yield. In this work, we take inspiration from a Mexican plant (*Echeveria pulvinate*) to prepare a material suitable for future water harvesting applications. Observation of *E. pulvinate* reveals that parahydrophobic properties are favorable for water harvesting. To mimic these properties, we leveraged a combination of 3D printing and post-functionalization to control surface wettability and obtain parahydrophobic properties. The prepared surfaces were investigated using IR and SEM. The surface roughness and wettability were also investigated to completely describe the elaborated surfaces and strongly hydrophobic surfaces with parahydrophobic properties are reported. This new approach offers a powerful platform to develop parahydrophobic features with desired three-dimensional shape.

## 1. Introduction

Water is a *sine qua none* condition for human development. Water is a key parameter critical to life, agriculture, and even hygiene. A lack of liquid water is one of the most significant problems facing a large portion of the global population. Due to climatic evolution, it has been estimated that in the near future (2025), a significant fraction of the global population will be in dramatic distress with regard to local water supply [1]. Such a scenario can lead to a catastrophic humanitarian situation, likely resulting in massive population migrations and related conflicts.

To avoid such an undesirable outcome, it is necessary to find a sustainable solution to supply water in these at-risk regions. For this reason, a significant body of research aimed at solving this problem and identifying solutions has been pursued. As one example, desalinization techniques that remove salt from seawater to produce drinkable water have been described [2,3,4,5,6]. This type of strategy is very efficient and can also produce a large quantity of water. This strategy is employed in Saudi Arabia, mainly for irrigation [7,8]. Despite the success of desalinization, this strategy presents a major drawback with regards to sustainability. Mainly, if salt is removed to yield water, what are we doing with the leftover salt? Desalinization produces significant quantities of salt, generating hyper-saline regions and salt deserts that sterilize the surrounding area [9,10]. Another solution used to supply water in desert regions is through water harvesting [11,12,13]. In many remote locations, if liquid water is lacking, atmospheric water can be abundant and harvested to yield liquid water. Fog harvesting is employed in Chile for irrigation, as one example, and it is clear that this strategy is compatible with sustainable development [14,15]. Water harvesting is also generally cheap, since fog is often harvested on a network of simple nets. The major drawback to fog harvesting is the poor associated efficiency; only a small portion of the available water is collected. A sustainable solution to increase harvesting efficiency while maintaining low cost is a topic of interest for many researchers. One approach to identify a solution is to observe living species effective at water harvesting to understand how this strategy has been optimized in nature. A bioinspired water harvesting device can be an interesting solution to provide liquid water naturally and at a low cost, without significant energy consumption. It is well reported that in arid regions, many species (animal and vegetal) developed techniques to harvest atmospheric water. As one example, the Namib desert beetle (*Stenocara gracilipes*) traps water on their legs and body to supply water [16,17,18]. Similarly, cacti can collect water with their pines and thus hydrate their growth [19,20]. In previous work, we also reported the possibility for Mexican plants such as *Echeveria pulvinata* to harvest water from the air [21]. The capability for *E. pulvinata* or cacti to harvest water is due to different parameters. First, plants effective at harvesting water increase their contact surface via macrostructures, such as pines or hairs for cacti and *E. pulvinata*, respectively. Secondly, these macrostructures have unique wettability that facilitates the collection of water from the air. More specifically, the surface of pines or hairs has a high affinity with water, thus trapping water droplets so they remain stuck to the surface. However, the surface of these macrostructures is also patterned with domains that are highly hydrophobic, this ultimately aids in water droplet collection as efficient transport of droplets can occur in these hydrophobic domains.

With these observations, it is obvious that to create similar functionalities and features it is critical to control both surface macrostructure and wettability. In the present work, we leverage 3D printing and post-functionalization to prepare surfaces inspired by *E. pulvinata*. The desired surface macrostructure is elaborated using a 3D-printing strategy. This choice is motivated by the fact that in the past decade 3D printing has been reported and explored as a powerful tool since it allows one to directly control and adapt the macrostructure of a surface to meet the needs of an application [22,23,24,25,26]. Furthermore, 3D printing allows us to easily prepare a variety of surfaces, with or without macroscopic features so the influence of these structures can be determined explicitly. Surface wettability, which is also a critical variable, is ultimately more difficult to control. The wettability of a surface depends on two key variables: surface energy and morphology, and many strategies manipulating these two variables to tailor surface wettability are presented in the research literature [27,28,29,30,31,32,33]. In the present work, we employ a chemical strategy to manipulate wettability, specifically a copper post-functionalization strategy with carboxylic acid. Prior works have demonstrated that this approach is effective for the preparation of highly hydrophobic surfaces on copper foils and plates [34,35,36,37]. In this study, we will use this approach to functionalize 3D-printed surfaces comprised of materials loaded with copper. With this combination of 3D printing and post-functionalization, we will be able to manipulate all critical parameters of the surfaces, mainly macrostructure and wettability (Figure 1). Here, the successful surface modification is confirmed using infrared spectroscopy (IR) and post-functionalized surfaces were characterized for size and roughness to determine the degree of microstructure modification due to the functionalization. Additionally, the surface wettability was also investigated via contact angle measurements.

## 2. Materials and Methods

### 2.1. Surface Elaboration

#### 2.1.1. Surface Printing

Surfaces were printed using a 3D printer (Stream 30 pro MK 2) from volumic 3D (Nice, France). The filament used for printing was Cuivre 80 Ultra (Volumic Ultra, Nice, France). This filament is made of polylactic acid (PLA) and loaded with 80% of colloidal copper. The nozzle used for printing is a 25 µm brass nozzle. The extrusion temperature was 195 °C and the reception plate temperature was 60 °C.

#### 2.1.2. Surface Oxidation

The surfaces were immersed in a solution of NaOH (2.5 M) and (NH_4_)_2_S_2_O_8_ (0.13 M) and then placed on a shaker at room temperature for 2 h. During this period, the colorless solution became blue, confirming the release of Cu^2+^ ions. After this period, the surfaces were washed three times in water and once in ethanol. After washing, the surfaces can directly be used for the next step without drying.

#### 2.1.3. Surface Functionalization

The oxidized surfaces were immersed in carboxylic acid solution in ethanol (10 mg·mL^−1^) and slowly agitated at room temperature for 2 h. After this period, the surfaces were washed three times in ethanol. The surfaces were then dried in an oven overnight at 70 °C.

### 2.2. Surface Characterization

#### 2.2.1. Electronic Microscopy

SEM (scanning electron microscopy) observations were carried out using Phenom Pro X Desktop SEM from Thermo Fisher Scientific (Waltham, MA, USA). Samples were observed with gold coating and at an accelerating voltage of 5 and 10 kV. The samples were coated using Quorum Q150R S Sputter Coater (Lewes, UK).

#### 2.2.2. FT-IR Characterization

Infrared measurements were carried out using a Spectrum Two FT-IR spectrometer from Perkin Elmer (Waltham, MA, USA) with universal ATR accessory. The measurements were performed between 4000 cm^−1^ and 500 cm^−1^.

#### 2.2.3. Contact Angle Measurement

All contact angle measurements were performed on non-functionalized and post-functionalized surfaces. All measurements were performed five times on three different samples to consider the standard deviation of measurements. The apparent contact angles were obtained with a contact angle system OCA TBU100 (dataphysics, Filderstadt, Deutshland). The apparent contact angles were measured using the sessile drop method with deionized water (droplet volume of 2 µL).

#### 2.2.4. Roughness Measurement

Roughness measurements were performed using an optical profilometer (InfiniteFocus G5 plus) from Brucker Alicona (Graz, Austria). All measurements were performed three times on three different samples to calculate the standard deviation.

## 3. Results

### 3.1. Surface Elaboration and Functionalization

The aim of the present work is to develop a novel strategy to easily prepare parahydrophobic surfaces suitable for water harvesting applications. To reach this goal, we leveraged the many advantages associated with 3D printing technology. Three-dimensional (3D) printing is an efficient method as many surfaces can be developed with varying geometries and the envisioned surfaces can be printed with commercially available materials at a relatively low cost. This strategy will allow us to use 3D-printed surfaces as a platform for additional post-functionalization. As a first trial, our selected morphology for printed surfaces was simple, a flat circle. To enable post-functionalization, a commercially available polylactic acid (PLA) loaded with 80% of colloidal copper we employed as the material for printing. This selection was motivated by the fact that copper is known to be suitable for functionalization using carboxylic acids. The post functionalization was performed following a method previously described in the literature [35]. In this two-step strategy, copper is initially oxidized and subsequently reacted with carboxylic acid (Figure 2).

Due to the wide diversity of commercially available carboxylic acids, this approach is a powerful pathway to manipulate surface wettability. To enable post-functionalization, the copper-loaded surface is first immersed in a solution of ammonium persulfate and sodium hydroxide. With this step, the copper is oxidized from Cu(0) to Cu(OH)_2_. The formation of Cu^2+^ is easy to observe due to the blue coloration of the Cu^2+^ solution (Figure 3).

After oxidation, the surface is washed three times with water and once with absolute ethanol. The wet surface is then immediately immersed in a carboxylic acid solution in ethanol. Here, we varied the carboxylic acid employed to obtain a range of surface wettabilities, from hydrophilic to hydrophobic. More specifically, different lengths of linear carboxylic acids were selected: hexanoic acid (C6), decanoic acid (C10), tetradecanoic acid (C14), and octadecanoic acid (C18). Additionally, benzoic acid (Benz), 4,4,5,5,6,6,7,7,8,8,9,9,9-tridecafluorononanoic acid (F), 2-(2-methoxyethoxy)acetic acid (Methoxy), and glycolic acid (Glycolic) were also selected. For each analog, 100 mg of carboxylic acid was dissolved in 10 mL of absolute ethanol. In the majority of the tested carboxylic acids, the solutions remained colorless (Figure 4A) but with the hexanoic, decanoic, and benzoic acids a slightly green/blue aspect was observed (Figure 4B).

After two hours in the carboxylic acid solution, the surface was removed and washed three times with ethanol. Lastly, the surface was dried overnight at 70 °C.

### 3.2. Infrared Measurements

After functionalization, the surfaces were initially investigated via infrared (IR) spectroscopy to confirm surface functionalization. Unfortunately, due to the limitation of the ATR-FTIR method, wavenumbers between 2800 cm^−1^ and 3700 cm^−1^ could not be considered [38]. Therefore, it was not possible to investigate OH, CH_2_, and CH_3_ stretching bands and thus we will focus on the carbonyl bands between 1700 cm^−1^ and 1750 cm^−1^ here. The IR spectrum from a non-modified surface revealed a well-defined band at 1730 cm^−1^. This band is attributed to the carbonyl of the ester group in PLA (Figure 5A).

After oxidation, the surface (Cu(OH)_2_) presented a band at 1730 cm^−1^ and also two shouldering bands at 1720 cm^−1^ and 1714 cm^−1^ (Figure 5B). This observation is consistent with partial degradation of the PLA due to the aggressive conditions of the functionalization. This degradation releases free carboxylic groups that can explain these new bands. Unfortunately, after surface post-functionalization via grafting of carboxylic acids, no change was observed in the IR spectra, as indicated by the IR spectra of C14 and Benz surfaces (Figure 5C,D, respectively). In both cases, three shouldering bands are observed near 1730 cm^−1^, 1720 cm^−1^, and 1714 cm^−1^. This observation was made for all studied surfaces with varying carboxylic acids (see Appendix A). While the IR analysis allows us to confirm the modification of the surfaces as compared to the starting material (Cu), it is not possible to prove the functionalization with this technique. Two explanations can be considered, first that the functionalization is only at the interface with Cu(OH)_2_ and as a consequence, this thin layer of grafted carboxylic acid is too weak to be observed via IR. Secondly, as all observable functional groups (carboxylic acid or ester) appear with similar wavelengths it is possible that they mask each other. As consequence, no difference was observed between samples prior to or after functionalization.

### 3.3. SEM Observations

All surfaces were investigated via SEM to determine microscopic surface morphology (Figure 6).

Low magnification SEM imaging of the non-modified surface (Cu) revealed a porous polymer with inclusions that can be attributed to the colloidal copper (Figure 6A). After oxidation (Cu(OH)_2_), the morphology was quite different with significant roughness and some exposed particles (Figure 6B). This result is consistent with the superficial degradation of PLA due to the functionalization conditions. This side effect is beneficial for our strategy, since the partial degradation of the surface partially frees the particles, thus increasing the reactive surface area of copper. After functionalization, no particular difference in surface morphology was observed between the oxidized surface (Cu(OH)_2_) and surfaces grafted with carboxylic acids, similar to the observations associated with the IR measurements. Representative images of methoxy and F surfaces are presented in Figure 6C,D (SEM images of all surfaces are presented in Appendix A).

At higher magnification, SEM revealed more information on the studied surfaces (Figure 7).

For the Cu (non-functionalized) surface, a relatively smooth surface was observed (Figure 7A). On the other hand, all modified surfaces presented rough features and a thin crystal network structure. This observation is consistent with the literature for this type of reaction [39,40]. Similar to the low magnification images, no major differences were observed between the oxidized surface and functionalized surfaces (Figure 7B–D). Only the surface modified with tetradecanoic acid (C14) presented a significantly different morphology (Figure 8).

Observed with low magnification, the C14 surface (Figure 8A) revealed similar surface morphology as the other surfaces modified with carboxylic acids (e.g., surfaces presented in Figure 7). However, at higher magnification, significantly different crystallization features were observed on the surface. Unlike other surfaces analyzed in this study, the crystals did not form a thin network, but instead presented features having the shape of sand roses (Figure 8B–D).

### 3.4. Surfaces Roughness

To complete surface characterization, the roughness of the surfaces was investigated. All surface roughness measurements with associated standard deviations are presented in Table 1. The data revealed that all surfaces had significant roughness with Ra between roughly 20 and 30 µm.

This roughness can be linked to the printing procedure and material employed. Printing was performed with a 25 µm nozzle, which is consistent with the observed roughness values. Globally, it is possible to observe that the surface’s roughness changes slightly during the functionalization. However, this change cannot be reported as significant due to the associated standard deviations. The large standard deviation is related to swelling and contraction that occurs during the functionalization steps (wetting and drying). In addition to roughness, shape measurements were also performed. This was carried out to compare the printed, modified surfaces and the informatically modelized shape and to confirm the preservation of the printed surface during the modification. The printed and modified shapes were measured for their sizes and reported in percentage to the theoretical size of the model. The obtained value is considered as the accuracy, and these values are reported in Table 2.

These observations reveal a modest loss in the size of the samples after modification, but the modified surfaces remain very similar to the modelized surface. The majority of modified surfaces have nearly 96–97% accuracy; these values are not perfect but remain acceptable. Overall, after post functionalization, the surface remained consistent with the theoretical model.

### 3.5. Surface Wettability

All surfaces were investigated for their wettability (Table 3). It is well established that surface wettability is linked to two main parameters: surface roughness and surface energy (chemical composition). As we have shown, all surfaces presented similar roughness. As consequence, variations in surface wettability should be linked to the surface chemical composition. As a reference, the non-modified Cu surface was investigated and found to have a hydrophobic character with an apparent contact angle *θ* = 101.1 ± 10.8°. After oxidation, the wettability was dramatically altered, and the surface Cu(OH)_2_ was highly hydrophilic (*θ* = 30.0 ± 7.0°). This change is consistent with both the formation of Cu(OH)_2_ and partial degradation of the PLA polymer which releases a polar group. As expected, due to the intrinsic hydrophobic feature of alkyl chains, surfaces modified with linear carboxylic acids became highly hydrophobic. Surfaces C6, C10, C14, and C18 presented apparent contact angles of *θ* = 134.1 ± 3.2°, *θ* = 147.9 ± 4.8°, *θ* = 145.4 ± 1.8° and *θ* = 145.3 ± 2.7°, respectively (Figure 9A).

The glycolic surface and methoxy surface presented apparent contact angles of *θ* = 31.5 ± 5.2° and *θ* = 66.8 ± 6.3°, respectively (Figure 9B). This change in wettability is due to the intrinsic hydrophilicity of these acids. Finally, the Benz and F surfaces presented highly hydrophobic behavior, with apparent contact angles of *θ* = 132.3 ± 3.3° and *θ* = 136.3 ± 4.9°, respectively. Since the surfaces developed here are intended for use in water harvesting, an important parameter is the parahydrophobic feature (petal effect). As observed for *E. pulvinata*, parahydrophobicity correlates with strong hydrophobic properties as well as strong adhesion of water on a surface. To determine if the developed surfaces are parahydrophobic, the surfaces upon which a water droplet has been deposited were tilted to an angle of 90° [28,41,42]. The observed behavior upon tilting revealed that all the prepared surfaces presented high adhesion with water, and thus can be reported as parahydrophobic (Figure 10).

With these results, it is evident that this strategy allows us to control the wettability of the printed surface and specifically to develop parahydrophobic surfaces. Given these results, it is also interesting to determine if this approach is compatible with more complex morphologies having more significant surface contact area (see examples, Figure 11).

The models presented in Figure 11A were designed and printed using copper-loaded PLA. One example is a circle with perpendicular blades intended to increase the contact surface of the structure. The second is a flower-inspired shape. The printed shapes were both functionalized following the procedure previously described (oxidation followed by functionalization). As this assay was completed to highlight the potential of this strategy for parahydrophobic surface elaboration, this experiment was carried out using one carboxylic acid, octadecanoic acid. Like the results for flat surfaces, it is possible to observe that after modification the complex structures became highly hydrophobic with parahydrophobic properties (Figure 11B–D), demonstrating the promising potential of the method. This proof of concept shows that this strategy is a powerful tool, suitable for original surface elaboration with water harvesting properties.

## 4. Conclusions

In this work, we report for the first time the combination of 3D printing and post-functionalization for parahydrophobic surface elaboration. Surface functionalization of 3D-printed substrates was explored following a two-step process and with a broad range of carboxylic acids. The functionalized surfaces were characterized using different techniques. While infrared spectroscopy did not allow us to confirm the reaction of the carboxylic acid with the oxidized surface, the modification between the Cu surface and the Cu(OH)_2_ surface was confirmed. Roughness and size measurements were conducted to confirm that the functionalization of the surface preserved the printed shape. Finally, the surface wettability was investigated. Our study reveals the significant impact of functionalization on surface properties. In particular, surfaces functionalized with linear carboxylic acids revealed a strong hydrophobic character with apparent contact angles near 140°. All the prepared surfaces exhibited parahydrophobic features (e.g., hydrophobicity and strong water adhesion), thus confirming the potential of this approach for water harvesting technology development. This strategy has also been applied for more complex printed structures to improve water harvesting potential. Future and ongoing work will evaluate the harvesting capacity of the elaborated surfaces.

## Figures and Tables

**Figure 1 biomimetics-06-00071-f001:**
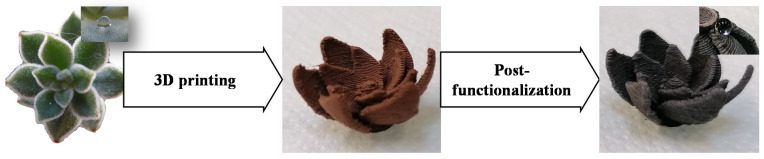
General concept of surface elaboration.

**Figure 2 biomimetics-06-00071-f002:**
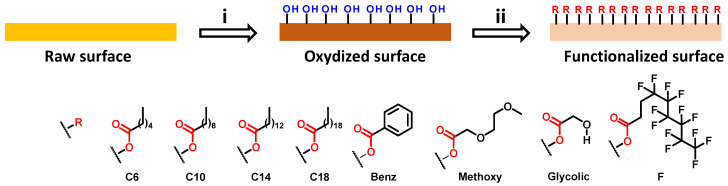
Mechanism of copper loaded surface post-functionalization. (**i**) Ammonium persulfate, sodium hydroxide, water, room temperature, 2 h. (**ii**) Carboxylic acid, ethanol, room temperature, 2 h.

**Figure 3 biomimetics-06-00071-f003:**
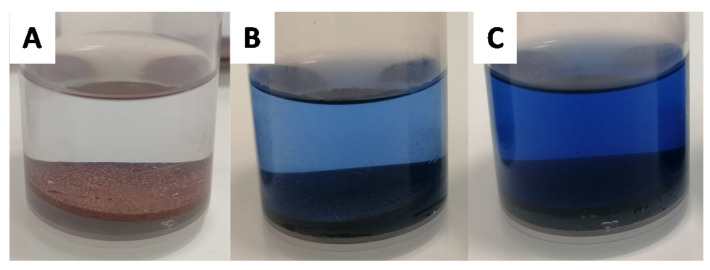
Coloration during oxidation step ((**A**): T = 0 h, (**B**): T = 1 h and (**C**): T = 2 h).

**Figure 4 biomimetics-06-00071-f004:**
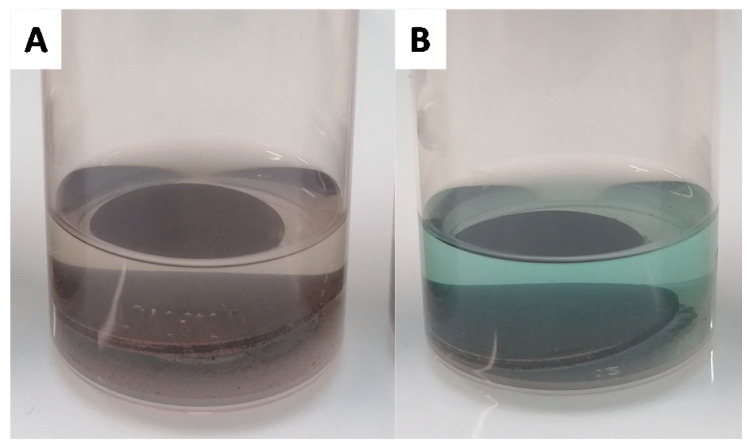
Functionalization with octadecanoic acid (**A**) and with hexanoic acid (**B**) after 2 h.

**Figure 5 biomimetics-06-00071-f005:**
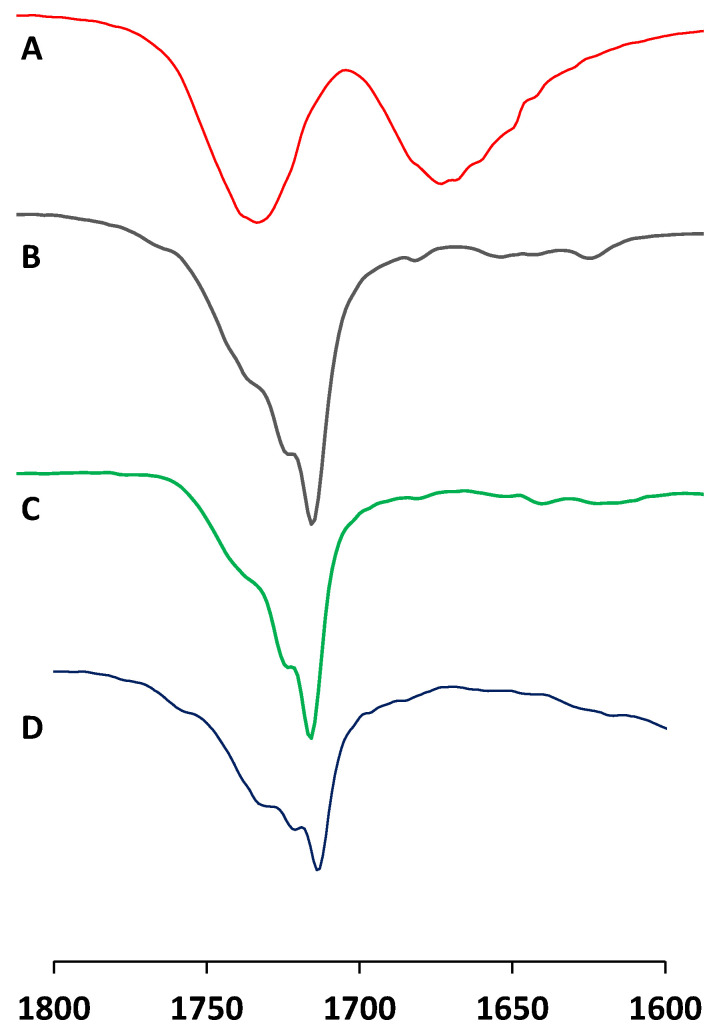
Examples of IR spectra for raw and modified surfaces. (**A**) Bare 3D-printed surface loaded with colloidal copper (Cu), (**B**) 3D-printed surface after oxidation (Cu(OH)_2_), (**C**) 3D-printed surface after functionalization with tetradecanoic acid (C14) and (**D**) 3D-printed surface after functionalization with Benzoic acid (Benz).

**Figure 6 biomimetics-06-00071-f006:**
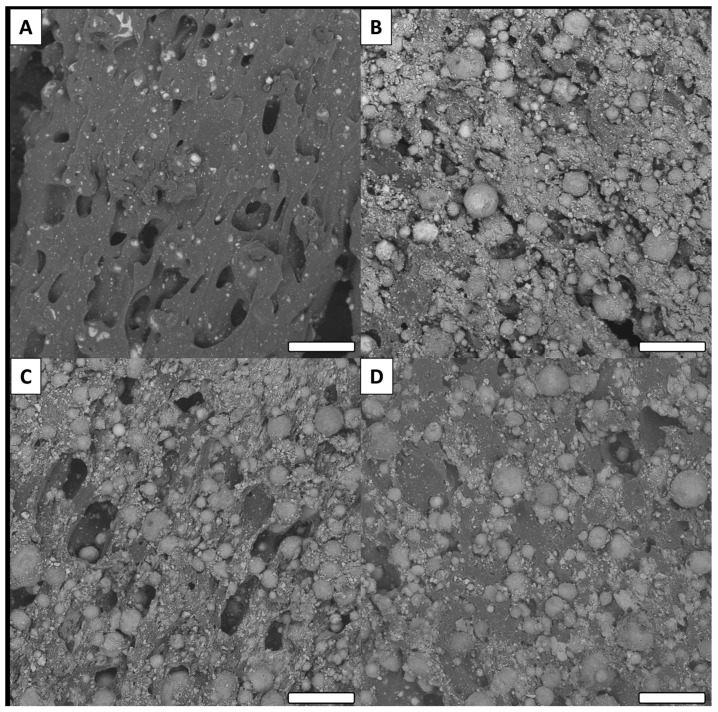
Examples of SEM images for raw and modified surfaces (low magnification). (**A**) Cu, (**B**) (Cu(OH)_2_), (**C**) methoxy and (**D**) F (scale bar = 100 µm for all images).

**Figure 7 biomimetics-06-00071-f007:**
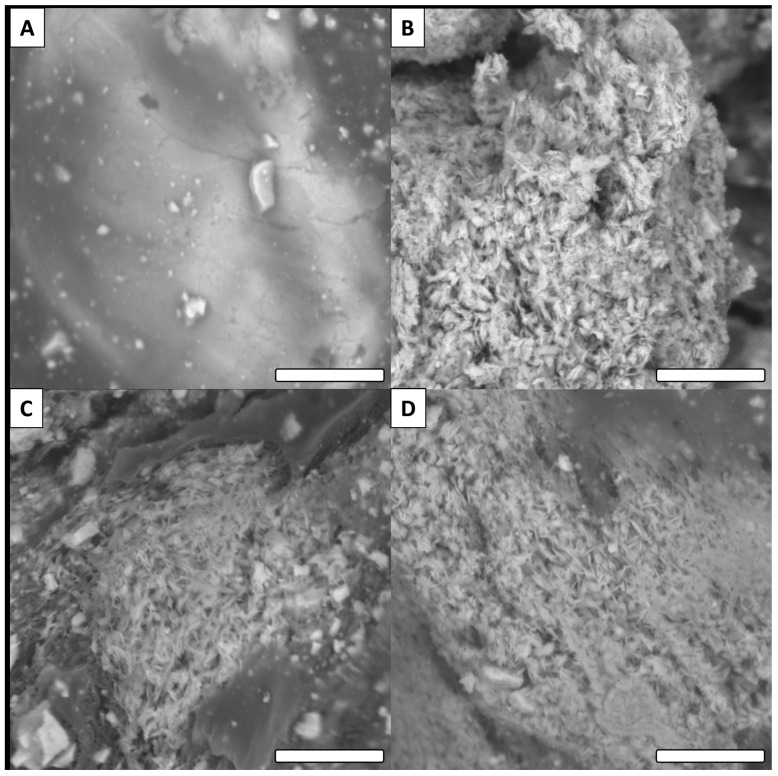
Examples of SEM images for raw and modified surfaces (high magnification). (**A**) Cu, (**B**) (Cu(OH)_2_), (**C**) methoxy and (**D**) F (scale bar = 8 µm for all images).

**Figure 8 biomimetics-06-00071-f008:**
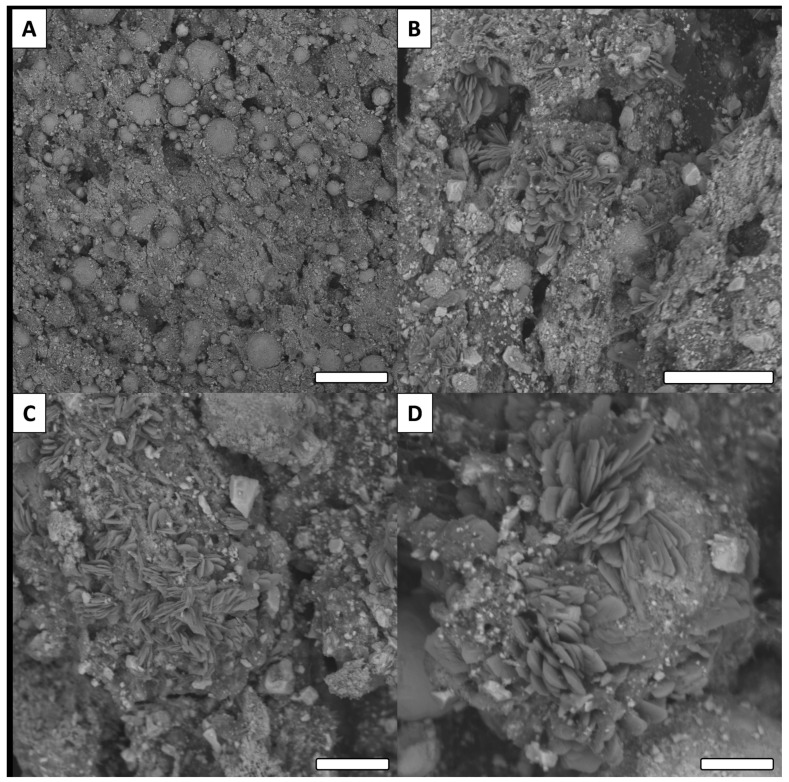
Examples of SEM images for C14 surfaces ((**A**): scale bar = 100 µm, (**B**): Scale bar = 30 µm, (**C**,**D**): Scale bar = 10 µm).

**Figure 9 biomimetics-06-00071-f009:**
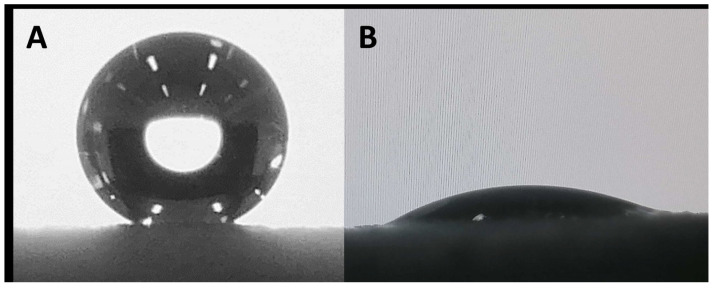
Example of water drop deposed on C10 surface (**A**) and glycolic surface (**B**).

**Figure 10 biomimetics-06-00071-f010:**
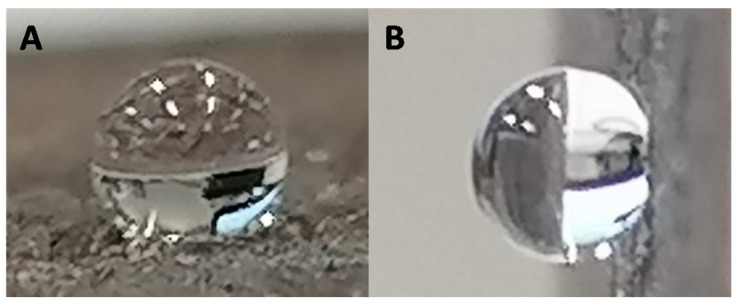
Example of parahydrophobic feature for C14 surface. (**A**). Water drop on horizontal surface. (**B**). Water drop on tilted surface.

**Figure 11 biomimetics-06-00071-f011:**
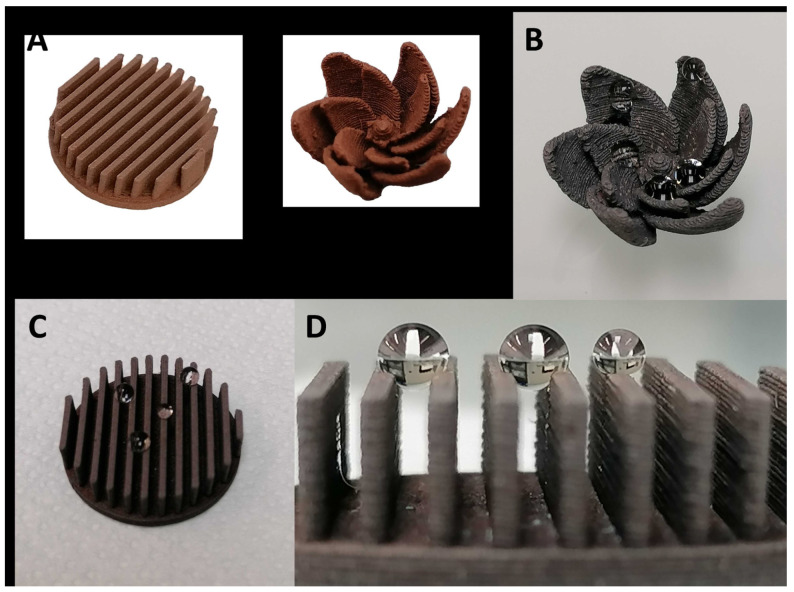
Examples of more complex surfaces printed with copper-loaded PLA and used for post-functionalization with octadecanoic acid. (**A**) New structures; (**B**) water deposited on flower structure functionalized with octadecanoic acid; (**C**,**D**) water deposed on circle with blade structures functionalized with octadecanoic acid.

**Table 1 biomimetics-06-00071-t001:** Roughness values for raw and modified surfaces. The presented Ra measurements are average values from five measurements from three examples of each sample ± standard deviation.

Sample	Ra (µm)
Cu (0)	19.5 ± 2.8
Cu(OH)_2_	23.5 ± 0.8
C6	24.4 ± 8.7
C10	30.4 ± 10.3
C14	25.8 ± 8.8
C18	33.7 ± 14.8
Glycolic	21.7 ± 9.2
Methoxy	33.2 ± 16.5
Benz	29.0 ± 13.8
F	20.9 ± 4.1

**Table 2 biomimetics-06-00071-t002:** Percentage of accuracy for the printed shapes compared to the theoretical model. The presented accuracy are average values from five measurements from three examples of each sample ± standard deviation.

Surface	Accuracy (%)
Cu	99.2 ± 0.9
Cu(OH)_2_	97.9 ± 1.2
C6	96.1 ± 1.1
C10	96.1 ± 0.7
C14	96.4 ± 0.4
C18	96.7 ± 1.3
Gly	96.7 ± 1.6
Methoxy	96.6 ± 1.0
Benz	95.8 ± 1.3
F	96.6 ± 0.8

**Table 3 biomimetics-06-00071-t003:** Wettability measurements for raw and modified surfaces. The presented apparent contact angles are average values from five measurements from three examples of each sample ± standard deviation.

Scheme 101	Apparent Contact Angle (°)
Cu	101.1 ± 10.8
Cu(OH)_2_	30.0 ± 7.0
C6	134.1 ± 3.2
C10	147.9 ± 4.8
C14	145.4 ± 1.8
C18	145.3 ± 2.8
Glycolic	31.5 ± 5.2
Methoxy	66.8 ± 6.3
Benz	132.3 ± 3.3
F	136.3 ± 4.9

## Data Availability

Not applicable.

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
