# Peer review of "Bioinspired and Post-Functionalized 3D-Printed Surfaces with Parahydrophobic Properties"

_biomimetics, 2021, doi:10.3390/biomimetics6040071_

Round 1

Reviewer 1 Report

Comments

Guilhem et al., reported a combination of 3D-printing and postfunctionalization strategy to control surface wettability and obtain parahydrophobic feature. The prepared surfaces were investigated using FT-IR and SEM. The surface roughness and wettability were also investigated to completely describe the elaborated surfaces and strongly hydrophobic surfaces with parahydrophobic behavior is reported. This new approach offers a powerful platform to develop parahydrophobic features with desired 3-dimensional shape. Therefore, this reviewer believes that minor revision is required to address the following comments before this manuscript can be considered for publication.

  1. I suggest adding references to Nano Today 2021, 39, 101182; Nature communications, 2020, 11, 1-8; Applied Materials Today 2020, 20, 100689 about the potential applications of 3D printing technology.
  2. XPS analysis was suggested to be added to demonstration of the copper oxidization.
  3. For SEM images, high resolution should be provided to check the detailed morphology change after different functionalization steps.
  4. The authors stated that the prepared surfaces exhibiting parahydrophobic features showed potential for water harvesting technology. However, no experiment was provided to demonstrate this point.
  5. More detailed Materials and Methods should be added, for example detailed parameters for surface printing.
  6. A detailed discussions were suggested to explain the concepet of “Bioinspired”, and its advantages for water harvesting.

Author Response

I suggest adding references to Nano Today 2021, 39, 101182; Nature communications, 2020, 11, 1-8; Applied Materials Today 2020, 20, 100689 about the potential applications of 3D printing technology.

Author's answer: The suggested references have been added.

XPS analysis was suggested to be added to demonstration of the copper oxidization.

Author's response: The group thanks the reviewer for his suggestion which is true, unfortunately our group is not equipped with an XPS device, and it is not possible to send the samples for analysis within the time limit of 5 days left for review.

For SEM images, high resolution should be provided to check the detailed morphology change after different functionalization steps.

Author's answer: SEM images of all functionalized surfaces has been provided with the highest available resolution. The document may have been compressed when converted as PDF. All images from main article and supplementary data will be provided with maximum resolution.

The authors stated that the prepared surfaces exhibiting parahydrophobic features showed potential for water harvesting technology. However, no experiment was provided to demonstrate this point.

Author's answer: The group agree with the reviewer comment, this comment was initially to introduce the prospective of our work. The authors reformulate the title to clarify this point.

More detailed Materials and Methods should be added, for example detailed parameters for surface printing.

Author’s answer: Details of the surface printing have been added to the materials and methods as recommended.

A detailed discussions were suggested to explain the concepet of “Bioinspired”, and its advantages for water harvesting.

Author’s answer: An explanation sentence has been added in the main text as suggested.

Reviewer 2 Report

    In the present manuscript, the authors reported a novel parahydrophobic surface elaboration which exhibited high hydrophobic features via 3D printing and post functionalization. The results are interesting and important for the potential applications in water colleting. I recommend it to be published in the journal of Biomimetics after the following corrections.

  1. As is known, the surface groups and its distribution are important to their surficial properties including hydrophobicity, more surface characterizations such as XPS and STEM should be supplied.
  2. How to understand the “all the prepared surface present high adhesion 283 with water even the more hydrophobic one” shown in Figure 12 in page 13?
  3. Some spelling mistakes could be found, and the English should be improved.

Author Response

As is known, the surface groups and its distribution are important to their surficial properties including hydrophobicity, more surface characterizations such as XPS and STEM should be supplied.

Author's answer: The group thanks the reviewer for his suggestion which is true, unfortunately our group is not equipped with an XPS device, and it is not possible to send the samples for analysis within the time limit of 5 days left for review.

How to understand the “all the prepared surface present high adhesion 283 with water even the more hydrophobic one” shown in Figure 12 in page 13?

Author’s answer: The authors thank the reviewer for his kind comment. The group rewrite this sentence to make it more understandable.

Some spelling mistakes could be found, and the English should be improved.

Author’s answer: To improve English, the manuscript has been double checked by a native speaker.
